# Inhibition of histone H3-H4 chaperone pathways rescues *C. elegans* sterility by H2B loss

**Ruixue Zhao**[1], **Zhiwen Zhu**[1], **Ruxu Geng**[1], **Xuguang Jiang**[1], **Wei Li**[2], **Guangshuo Ou**[1]*

**1** Tsinghua-Peking Center for Life Sciences, Beijing Frontier Research Center for Biological Structure, McGovern Institute for Brain Research, School of Life Sciences and MOE Key Laboratory for Protein Science, Tsinghua University, Beijing, China, **2** School of Medicine, Tsinghua University, Beijing, China

* guangshuoou@tsinghua.edu.cn

**Data Availability Statement:** All data is available in the main text and its Supporting Information files. All Chip-Seq sequencing data from this study have been submitted to Gene Expression Omnibus database (GEO) under accession number

## Abstract

Oncohistone mutations are crucial drivers for tumorigenesis, but how a living organism governs the loss-of-function oncohistone remains unclear. We generated a histone H2B triple knockout (3KO) strain in *Caenorhabditis elegans*, which decreased the embryonic H2B, disrupted cell divisions, and caused animal sterility. By performing genetic suppressor screens, we uncovered that mutations defective in the histone H3-H4 chaperone UNC-85 restored *H2B 3KO* fertility by decreasing chromatin H3-H4 levels. RNA interference of other H3-H4 chaperones or H3 or H4 histones also rescued *H2B 3KO* sterility. We showed that blocking H3-H4 chaperones recovered cell division in *C. elegans* carrying the oncohistone H2B$^{E74K}$ mutation that distorts the H2B-H4 interface and induces nucleosome instability. Our results indicate that reducing chromatin H3-H4 rescues the dysfunctional H2B in vivo and suggest that inhibiting H3-H4 chaperones may provide an effective therapeutic strategy for treating cancers resulting from loss-of-function H2B oncohistone.

## Author summary

Dysfunctional histones cause developmental defects and drive tumorigenesis. Using *Caenorhabditis elegans* to study histone H2B loss, we found that inhibiting histone H3-H4 chaperones restored fertility in H2B triple knockouts by decreasing chromatin H3-H4 levels. The blockade of H3-H4 chaperones recovered cell division in *C. elegans* carrying the oncohistone H2B$^{E74K}$ mutation that distorts the H2B-H4 interface and induces nucleosome instability. We suggest that the overall stoichiometry of histones within nucleosomes is more important than the actual abundances of histone dimers or tetramers and that inhibiting H3-H4 chaperones may provide an effective therapeutic strategy for treating cancers resulting from loss-of-function H2B oncohistone.

GSE185582 (https://www.ncbi.nlm.nih.gov/geo/query/acc.cgi?acc=GSE185582).

**Funding:** G.O. received the findings from the National Natural Science Foundation of China (grants 31991190, 31730052, 31525015, 31861143042, 31561130153, 31671444 and 31871352) and National Key R&D Program of China (2017YFA0503501, 2019YFA0508401, and 2017YFA0102900). The funders had no role in study design, data collection and analysis, decision to publish, or preparation of the manuscript.

**Competing interests:** The authors have declared that no competing interests exist.

## Introduction

Recent efforts to catalog histone mutations in cancer patients have identified many histone missense mutations as oncogenic drivers or contributors to tumor progression [1,2]. Histone globular domains form the core of the nucleosome, and the unstructured histone tails provide the target sites for various posttranslational modifications (PTM) [2–4]. The best-characterized oncohistone mutations distribute in the histone tail, including H3$^{K27M}$ in diffuse intrinsic pontine gliomas, H3$^{G34V/R}$ in pediatric glioblastomas, or H3$^{K36M}$ in chondroblastomas [1,5,6]. In contrast to the well-known PTM-associated mutations, the most common oncohistone mutations localize in the globular histone fold domain, disrupting the structural integrity of the nucleosome [2,7]. Nucleosome assembly involves depositing a histone (H3-H4)$_2$ tetramer and rapidly following H2A-H2B heterodimer deposition [4]. The E76 residue of H2B interacts with the R92 residue of H4 through hydrogen bonding, and the H2B$^{E76K}$ oncohistone mutation profoundly destabilizes the interaction of H2B with H4 and inhibits histone octamer formation [2,7]. Despite the most frequent H2B oncohistone mutation, how a living organism controls H2B$^{E76K}$ remains elusive, and neither can we propose therapeutic strategies to treat H2B$^{E76K}$ tumors.

Histone chaperone pathways deliver nascent histone subunits from the cytoplasm to the nucleosome [8]. After histone synthesis, the somatic nuclear autoantigenic sperm protein (sNASP) presents newly folded histone H3-H4 dimers to acetyltransferase 1 HAT1 holoenzyme. The anti-silencing factor 1 (ASF1) family proteins shuttled the acetylated H3-H4 dimers into the nucleus. For replication-coupled nucleosome assembly, the H3-H4 dimer in the Asf1-H3-H4 complex is transferred to the CAF-1 complex to form the (H3-H4)$_2$ tetramer, which deposits onto newly synthesized DNA. To deposit replication-independent histone variants, ASF1 interacts with the HIRA complex to hand over H3.3-H4 to assemble the nucleosome [9]. The FACT complex transfers the H3-H4 tetramers onto DNA, whereas FACT and NAP1 deposit the H2A-H2B onto H3-H4 tetramers [10]. Deposition of H3.3-H4 at the pericentromeric and telomere regions requires the chaperone DAXX-ATRX [10]. Whether and how histone chaperones are involved in oncohistone regulation are unknown.

Functional redundancy is a significant technical hurdle for the genetic dissection of histone. A metazoan genome usually encodes 10–20 copies of the genes for each of the four core histone proteins [11,12]. We examined a complete inventory of green fluorescence protein (GFP)-tagged H2B knock-in *C. elegans* and uncovered four out of 17 H2B genes expressed in the germline and early embryos. By generating a series of H2B mutant alleles, we showed that deleting three of these four H2B genes (*H2B 3KO*) caused animal sterility with full penetration, which enabled us to perform forward genetic screens to isolate suppressors. Unexpectedly, we found that inhibiting H3-H4 histone chaperones or reducing H3-H4 histones restored the fertility of H2B 3KO animals. We provided evidence that the loss of H3-H4 histone chaperones rescued cell division failures in *C. elegans* bearing the H2B$^{E76K}$ mutation, suggesting that H3-H4 histone chaperones may be promising therapeutic targets for treating cancers resulting from dysfunctional H2B oncohistone mutations.

## Results

### UNC-85 mutations restore the fertility of H2B triple knockout *C. elegans*

A metazoan genome usually encodes 10–20 copies of the genes for each histone protein [11,12]; the functional redundancy has been a significant technical hurdle for the genetic dissection of histone genes in animal cells. In *C. elegans*, 17 annotated H2B genes encode five distinct polypeptides (Figs 1A and S1) [12]. By examining a complete inventory of green

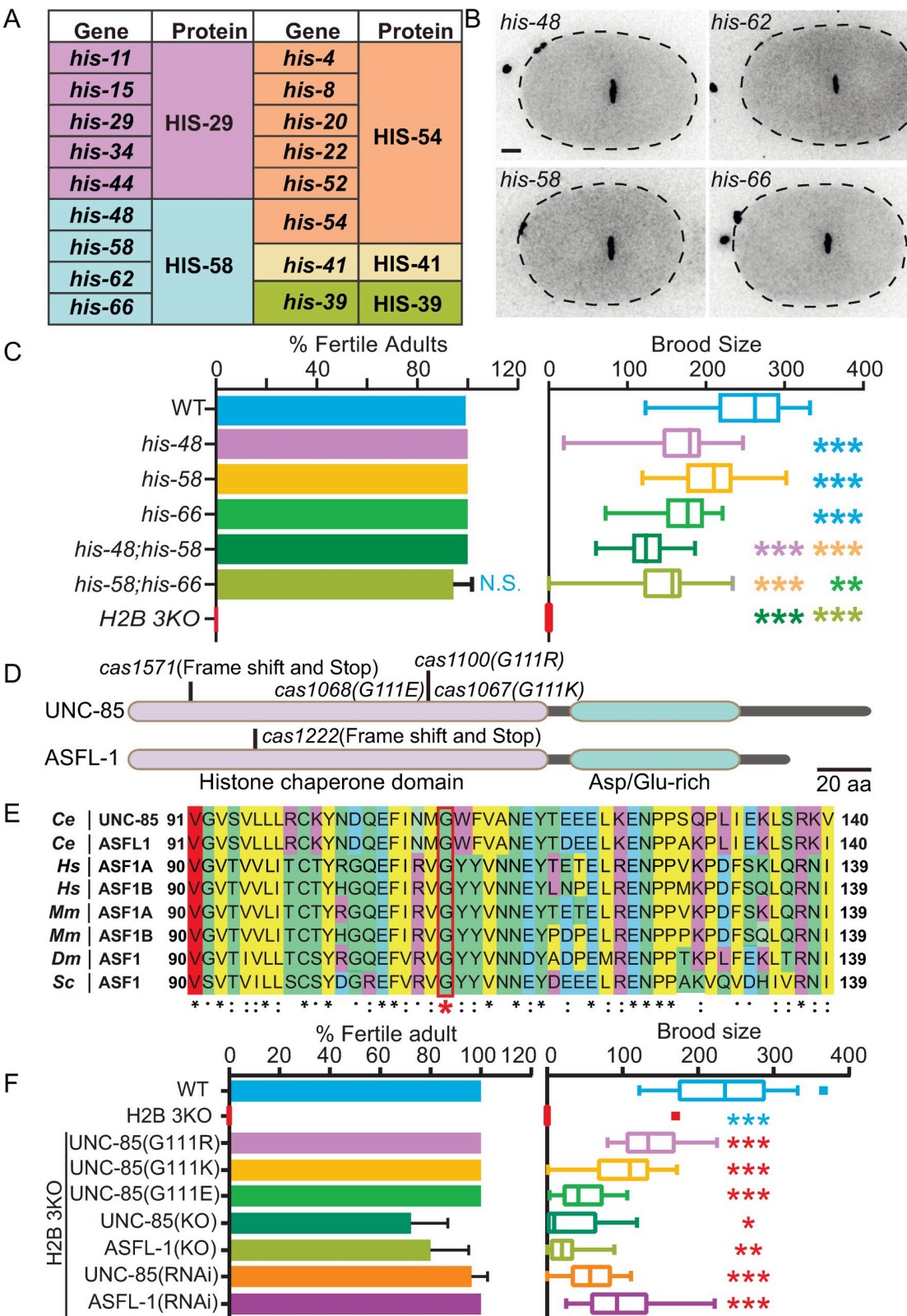

**Fig 1. UNC-85 mutations restore the sterility of *H2B 3KO C. elegans*.** (A) *C. elegans* H2B genes and proteins. (B) Localization of the endogenously GFP::tagged HIS-58 on chromosomes during metaphase in wild-type (WT) one-cell embryos. Scale bar, 5 μm. (C) Loss of H2B decreased brood size. N = 20–53, data are presented as mean ± SD (error bars). Statistical significance compared with

the control with a matching color code is based on Student's *t*-test, n.s., not significant, $^{**}p < 0.01$, $^{***}p < 0.001$. Three biological replicates calculated the fertile adult percentage. (**D**) Schematic of the *C. elegans* UNC-85 and ASFL-1 proteins. The amino acid changes (*cas1067, cas1068, and cas1100*) and the deletion allele of *unc-85 (cas1571)* or *asfl-1 (cas1222)* are indicated. The scale bar represents 20 amino acids. (**E**) Multi-sequence alignment of *C. elegans (Ce)* UNC-85 with ASFL-1 and its homologs in *human (Hs)*, *mouse (Mm)*, *Drosophila (Dm)*, and *Saccharomyces (Sc)*. The sequence alignment shows the protein sequence of UNC-85 histone chaperone domain flanking (± 20 aa) around the mutation isolated from the EMS screen, and the histone chaperone domain is highly acidic, which favors its interaction with histone. The red asterisk highlights the mutation isolated from the EMS screen. An asterisk (*) indicates positions 100% conserved in the alignment; a colon (:) indicates conservation between groups of strongly similar properties; a period (.) indicates conservation between groups of weakly similar properties. Hydrophobic residues are colored in blue, positive charge residues in yellow, negative charge residues in magenta, and polar residues in green. (**F**) Fertile adult percentage and brood size of the indicated strains. N = 10–70, values are presented as mean ± SD (error bars). Statistical significance compared with the control with a matching color code is based on Student's *t*-test, n.s., not significant, $^{*}p < 0.05$, $^{**}p < 0.01$, $^{***}p < 0.001$. Three biological replicates calculated the fertile adult percentage.

fluorescence protein (GFP)-tagged H2B knock-in (KI) strains, we found that the *C. elegans* germline and early embryos only expressed four H2B genes encoding the HIS-58 subfamily protein (Figs 1B and S2 and S3). To study the four germline and embryonic H2B genes, we generated the single, double, or triple H2B knockout (*H2B 3KO*) strains (S4A and S4B Fig). Immunofluorescence detected an expected H2B reduction in H2B 3KO embryos (S4C and S4D Fig). The loss of H2B significantly decreased brood sizes in the single or double H2B KO strains, and the *H2B 3KO* animals induced the sterility phenotype with a 100% penetration (Fig 1C).

We conducted forward genetic screens for suppressors that restore fertility in *H2B 3KO* (S5A Fig). Our genetic mapping, followed by whole-genome sequencing, identified the *unc-85 (cas1100)* mutant as a suppressor (Fig 1D–1F). The gene *unc-85* encodes one of two *C. elegans* homologs of anti-silencing function 1 (Asf1) histone H3-H4 chaperone crucial for chromatin assembly [3,9,13]. The *unc-85 (cas1100)* allele alters a conserved glycine at residue 111 to arginine, mapped to the histone chaperone domain (Fig 1D and 1E). Using the structural information of the yeast ASF1, we showed that the UNC-85$^{G111}$/Asf1$^{G110}$ localizes in β strand 7, a binding interface to H3-H4 histones (S5B and S5E Fig) [14]. Molecular dynamic simulation suggests that G111R mutation may perturb the interaction between the chaperone and H3-H4 (S5B–S5G Fig) and impair UNC-85 function. To understand the cellular behaviors of UNC-85$^{G111R}$, we compared the green fluorescence from the UNC-85::GFP and UNC-85$^{G111R}$::GFP KI animals. We did not detect a noticeable change in the UNC-85::GFP$^{G111R}$ signal in the whole cell; however, the nucleus-to-cytoplasm ratios of UNC-85::GFP were consistently higher than that of UNC-85::GFP$^{G111R}$ in the early embryos or oocytes (Figs 2A and 2B and S6A and S6B and S1 Video), indicating that G111R mutation reduces the nuclear localization of UNC-85.

We further determined the importance of the G111 residue by generating another two charge-altered mutations, G111K(*cas1067*) and G111E(*cas1068*). UNC-85$^{G111K}$ and UNC-85$^{G111E}$ rescued 44.1% and 20.7% of the WT brood size, respectively (Fig 1F). By creating a putative null allele of *unc-85(cas1571)* (Fig 1D), we confirmed that the loss of UNC-85 rescued sterility in *H2B 3KO* animals. In line with the mutant results, *unc-85* knockdown by RNA interference (RNAi) revealed sterility rescue effects on *H2B 3KO* (Fig 1F). By generating a deletion allele of the other *C. elegans* Asf1 H3-H4 chaperone *asfl-1(cas1222)* or using *asfl-1* RNAi, we showed that the loss of ASFL-1 restored the fecundity of *H2B 3KO* animals (Fig 1F). Collectively, these findings indicate that inhibiting histone H3-H4 chaperone UNC-85 and ASFL1 restored the animal viability and fertility of *H2B 3KO*.

## UNC-85 mutations rescue the meiosis defects of *H2B 3KO* animals

We followed spindle and chromosome dynamics to dissect the underlying cellular mechanism using GFP-tagged tubulin and mCherry-tagged histone H3 (HIS-72) in the *C. elegans* embryos

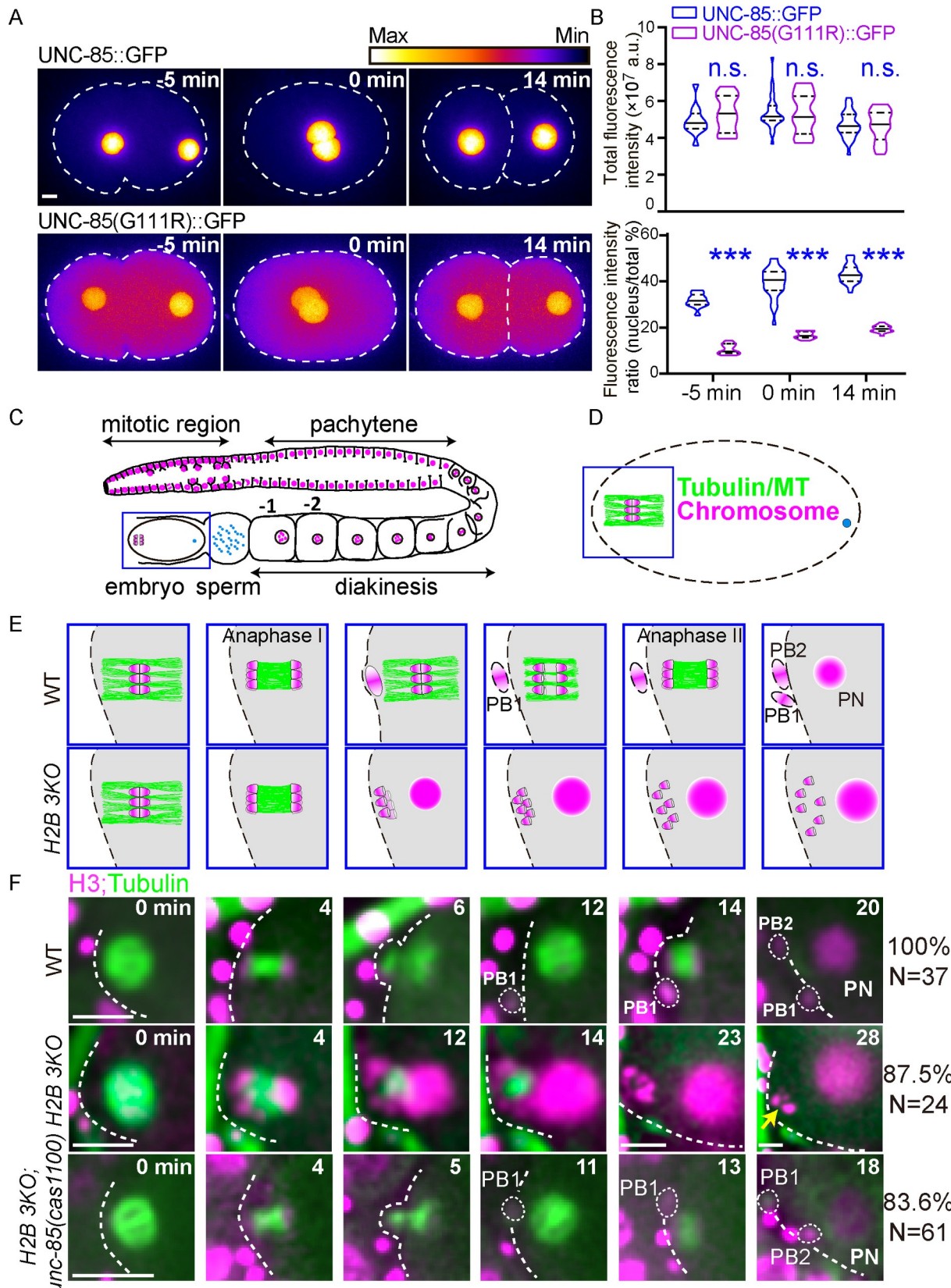

**Fig 2. UNC-85 mutations restore the meiosis defects of *H2B 3KO* animals.** (**A**) Fluorescence time-lapse heatmaps of UNC-85::GFP (up) and UNC-85::GFP$^{G111R}$ (down) in one-cell stage embryos. The first occurrence of pronucleus meeting was defined as time 0 min. Scale bar, 5 μm. (**B**) Quantifications of the whole-cell GFP-tagged UNC-85 fluorescence intensity (up) and intensity ratio of the nucleus to the cytoplasm (down). Data are presented as mean ± SD (error bars), N = 11–34. Statistical significance is based on two-way ANOVA, n.s., not significant, ***$p < 0.001$. (**C**) Schematic representation of an adult hermaphrodite gonad. (**D**) Schematic depiction of the fertilized embryo. MTs and chromosomes are indicated by green and magenta, respectively. (**E**) Enlarged schematic timescales of oocyte meiosis in WT (up) and *H2B 3KO* (down). (**F**) In utero fluorescence time-lapse images of meiosis visualized by GFP::tubulin (green) and mCherry::Histone (magenta) during oocyte meiosis in WT, *H2B 3KO* and *H2B 3KO; unc-85(cas1100)* mutant. Metaphase I was time zero. N (N = 24–61 animals, oocytes are from different animals) and and percentages of meiosis patterns were indicated on the right. Scale bar, 5 μm.

(Fig 2C and 2D). The wild-type (WT) embryos undergo two consecutive chromosome segregation events (i.e., meiosis I and meiosis II) preceded by a single round of DNA replication. As the spindle microtubules (MTs) gradually decreased, inter-chromosomal MTs assembled between segregating homologous chromosomes, followed by polar body exclusion. The second meiotic spindles were similarly formed (Fig 2E and 2F and S2 Video) [15]. In *H2B 3KO* animals, when the oocyte underwent meiotic maturation, meiotic spindle assembly and anaphase I correctly occurred; however, the spindle midzone persisted, the first polar body did not form, and the meiosis II spindle failed to assemble (Fig 2E and 2F and S2 Video), leading to an ectopic replicative fate and embryonic lethality. In the *H2B 3KO* and *unc-85 (cas1100)* quadruple mutant animals, 83.6% of the examined oocytes completed meiosis and mitosis successfully (Fig 2F and S3 Video). These results show that the loss of the H3-H4 chaperone UNC-85 rescued cell division failures in *H2B 3KO* animals.

## Inhibition of other components in histone H3-H4 chaperone pathways suppresses the sterility in *H2B 3KO*

Histone chaperone pathways deliver nascent histone subunits from the cytoplasm to the nucleosome [8], and additional histone chaperones relay with ASF1 to transport H3-H4 and facilitate their deposition to chromatin (Fig 3A) [9,16]. Next, we performed an RNAi screen to determine whether the loss of other histone chaperones rescues *H2B 3KO* sterility. After histone synthesis, the somatic nuclear autoantigenic sperm protein (NASP) presents newly folded histone H3-H4 dimers to acetyltransferase 1 HAT1 holoenzyme. Consistent with *unc-85* and *asfl-1* results, RNAi of *nasp-1* or *nasp-2*, the orthologs of the human NASP, efficiently suppressed sterility of *H2B 3KO* animals by partially restoring the brood size (Fig 3B). After ASF1 family proteins shuttled the acetylated H3-H4 dimers into the nucleus, the H3-H4 dimer in the Asf1-H3-H4 complex is transferred to the CAF-1 complex to form the (H3-H4)$_2$ tetramer, which deposits onto newly synthesized DNA. To deposit replication-independent histone variants, ASF1 interacts with the HIRA complex to hand over H3.3-H4 to assemble the nucleosome [9], and the chaperone DAXX-ATRX deposits H3.3-H4 at the pericentromeric and telomere regions [10]. We showed that RNAi of *chaf-1* (the human CHAF1A) or *hira-1* or *dap-1* (the human HIRA and DAXX homologs) rescued brood size defects in *H2B 3KO*. Our data revealed that inhibition of the H3-H4 chaperones with distinct functions rescued the sterility of *H2B 3KO*. Like many other species, the *C. elegans* early embryos do not undergo active gene transcription until the eight-cell stage, excluding the possibility that these chaperones play distinct regulatory roles in gene expression. Instead, we suspect that the blockade of any chaperone will reduce the H3-H4 amount from chromatin, restoring the overall stoichiometry of histones within nucleosomes close to the normal level. By contrast, RNAi of H2A-H2B chaperone NAP1 or FACT orthologs *nap-1* or *spt-16* did not suppress the sterility phenotype (Fig 3B). Thus, reducing components in the H3-H4 but not H2A-H2B chaperone pathways recovered the fertility of *H2B 3KO* animals.

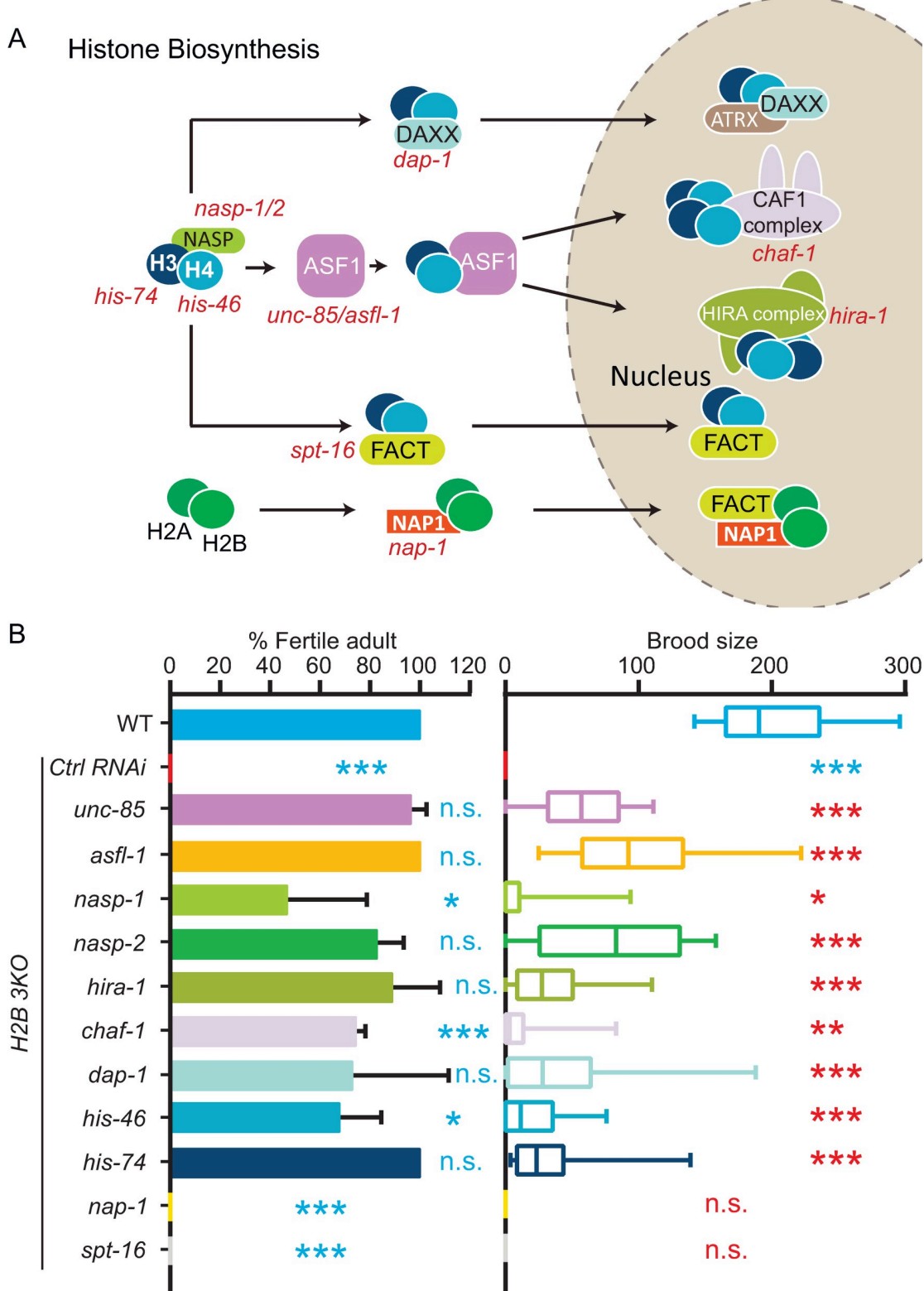

**Fig 3. RNAi of other components in H3-H4 chaperone pathways rescued *H2B 3KO* sterility. (A)** A model of the histone supply network. (**B**) Fertile adult percentage and brood size of *H2B 3KO* in the presence of control dsRNA (L4440) and dsRNAs targeting different histone chaperones. Data are presented as mean ± SD (error bars), N = 18–52. Statistical significance compared with the control with a matching color code is based on Student's *t*-test, n.s., not significant, $^*p < 0.05$, $^{**}p < 0.01$, $^{***}p < 0.001$. Three biological replicates calculated the fertile adult percentage.

## H3-H4 to H2A-H2B ratio is essential for cell divisions

How did the loss of H3-H4 chaperones restored fertility by the lack of H2B? Immunofluorescence of *H2B 3KO* and *unc-85* quadruple mutant embryos did not reveal any increase in H2B compared to *H2B 3KO* embryos (S4C and S4D Fig), excluding the possibility of H2B recovery by inhibiting H3-H4 chaperones. Because the quadruple mutant embryos are viable but have a low H2B level similar to that in *H2B 3KO* embryos (S4C and S4D Fig), we argue that H2B reduction may not be the primary cause for *H2B 3KO* sterility. To determine whether the loss of H3-H4 chaperone reduced histone H3 proteins in the nucleus, we constructed the HIS-74 (H3)::GFP KI reporter. The whole-cell HIS-74::GFP protein fluorescence in the *H2B 3KO* and *unc-85(cas1100)* quadruple mutant strain was indistinguishable from WT in early embryos or oocytes (Figs 4A and 4B and S7A and S7B and S4 Video). However, the nucleus-to-cytoplasm ratio of H3 fluorescence intensity significantly decreased, indicating a reduction of H3 recruitment to the nuclei, consistent with the established role of UNC-85 as an H3-H4 chaperone. To explore the histone H3 content on chromatin, we generated genome-wide maps for H3 chromatin immunoprecipitation with the exogenous reference genome (ChIP-Rx), followed by massively parallel sequencing [17]. Compared to WT, we detected a marked reduction in the widespread distribution of the total H3 in the *H2B 3KO* and *unc-85(cas1100)* quadruple mutant animals (Figs 4C–4E and S7C). We performed immunostaining experiments to examine the histone H4 levels using an anti H4 antibody that recognizes all the 16 *C. elegans* histone H4 proteins with an identical amino acid sequence. We found that H2B and H4 protein levels were markedly reduced in the viable quadruple mutant embryos (S7D and S7E Fig), suggesting that the aberrant H2A-H2B to H3-H4 ratio might be responsible for the sterility in *H2B 3KO*. In support of this idea, RNAi of the H3 *his-74* gene or H4 *his-46* gene in *H2B 3KO* animals restored brood size to 16.3% or 9.8% of WT (Fig 3B), respectively. These results collectively show that reducing the chromatin H3 or H4 level rescued the *H2B 3KO* sterility.

To disrupt the H2A-H2B to H3-H4 balance alternatively, we used a heat-shock inducible system to express H2B or H3 in the embryos under the control of the *Phsp-16.41* promoter (S8A Fig). The heat-shock induction led to a higher frequency of chromosome segregation defects, including anaphase bridges and lagging chromatin, and the embryos overexpressing H2B or H3 exhibited embryonic lethality with 38.2% or 52.4% penetrance, respectively (S8B–S8D Fig). Using the heat shock overexpression system, we observed that H2B or H3 overexpressed *C. elegans* embryos underwent abnormal chromatin segregation and defective cell division, consistent with the early results that overexpressed histone genes disrupted cell division in yeast and human cells [18–20]. Although we observed markedly higher penetrance of cell division failure in embryos overexpressing histone than the wild-type embryos treated by heat shock, we cannot exclude the effects of heat shock on embryonic development because heat shock of WT embryos caused 10~20% embryonic lethality [21,22]. These findings are consistent with the early yeast histone overexpression results [18] and support the notion that the proper H2A-H2B to H3-H4 ratio is essential for organizing chromatin and cell division.

## Inhibition of H3-H4 chaperones rescues abnormalities caused by H2B oncohistone in *C. elegans*

We wondered whether inhibiting H3-H4 chaperones could rescue cellular abnormalities caused by the loss-of-function H2B$^{E76K}$ oncohistone mutation [1,2]. We generated a *C. elegans* strain that changed the germline H2B *his-48(cas1172)* to E74K, equivalent to the human E76K (Fig 5A and 5B). The *his-58(cas1239)* and *his-66(cas1240)* double knockouts (H2B 2KO) or HIS-48$^{E74K}$ animals were fertile (Fig 5C); however, introducing HIS-48$^{E74K}$ into H2B 2KO decreased brood size and caused embryonic lethality in a temperature-dependent manner (Fig

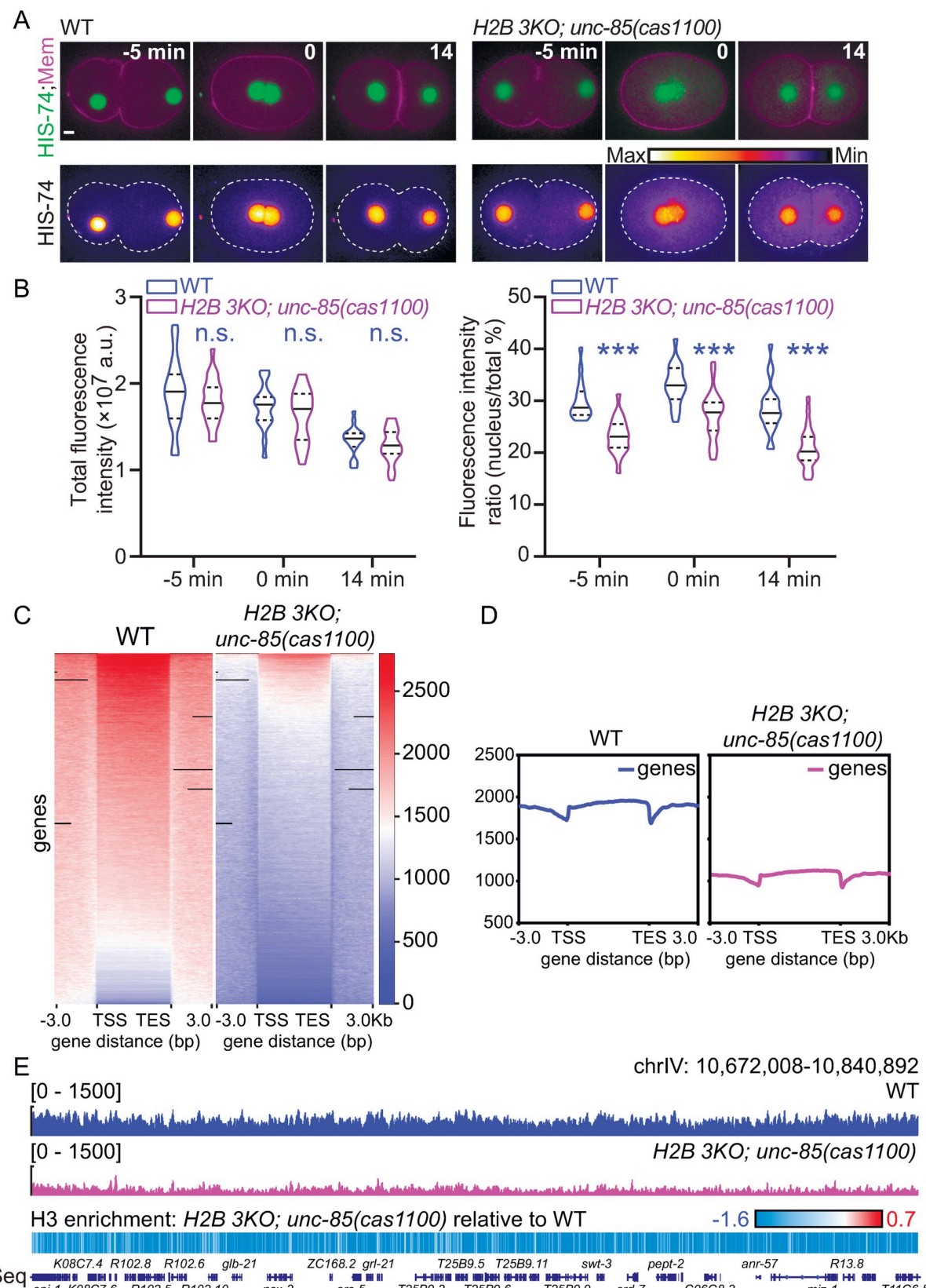

**Fig 4. The *unc-85* mutation reduces chromatin H3.** (A) Fluorescence time-lapse images (up) and heatmaps (down) of endogenously GFP-tagged HIS-74 (H3) with the mCherry-tagged (magenta) membrane during one-cell stage *C. elegans* embryo in WT (left) or *H2B 3KO; unc-85(cas1100)* quadruple mutant (right) embryos. Scale bar, 5 μm. (B) Quantifications of the whole-cell GFP-tagged HIS-74 fluorescence intensity (left) and intensity ratio of the nuclear to the cytoplasm (right). Data are presented as mean ± SD (error bars), N = 21–34. Statistical significance is based on two-way ANOVA, n.s., not significant, ***$p < 0.001$. (C) Heat maps show the signals of H3 flanking (± 3 kb) around the gene bodies. (D) Average enrichment profile of H3 reads flanking (± 3 kb) around the gene bodies; the y-axis represents the average normalized number of fragments at the corresponding genomic regions indicated in the x-axis. (E) Genome tracks show H3 signals of a representative genomic locus (Chr IV: 10,627,008–10,840,892) in WT (up) or *H2B 3KO; unc-85(cas1100)* quadruple mutants (middle). Heatmap shows H3 depletion of *H2B 3KO; unc-85(cas1100)* quadruple mutant relative to WT (bottom). Dark blue boxes show the reference genes.

5C and 5D). Although the embryos completed oocyte meiosis I and assembled meiotic II spindle, their chromosomes appeared more dispersed with the spindle relative to WT during anaphase II. The spindle midzone persisted and failed to extrude the polar body at the end of meiosis II (Fig 5E and 5F and S5 Video), generating multiple maternal pronuclei that disrupted the subsequent mitotic divisions (S8E Fig). By genetically crossing *unc-85 (cas1100)* or *asfl-1 (cas1222)* into the strain carrying H2B 2KO and H2B$^{E74K}$, we found an increased brood size and a reduction of embryonic lethality (Fig 5C and 5D and S6 Video). Live-cell imaging analyses showed that *unc-85 (cas1100)* successfully recovered meiosis II and subsequent mitoses in the embryos (Fig 5E and S6 Video), indicating the loss of H3-H4 chaperones rescues the defects caused by the dysfunctional oncohistone H2B$^{E74K}$ in *C. elegans*. We did not notice that the single HIS-48$^{E74K}$ mutation causes apparent sterility (Fig 5C), likely due to the functional redundancy of histone genes. We speculate that the other three wild-type H2B genes generate normal H2B proteins that sustain embryo development in the *his-48$^{E74K}$* single mutant. However, if we deleted another two H2B genes, i.e., *his-58$^{KO}$* and *his-66$^{KO}$*, the only left WT H2B gene may not produce a sufficient amount of normal H2B protein to maintain the proper chromatin structure that supports early development in HIS-48$^{E74K}$, *his-58$^{KO}$* and *his-66$^{KO}$* triple mutant embryos. Besides, we deleted two germline H2B genes and changed the other two H2Bs into H2B$^{E74K}$, and *unc-85 (cas1100)* did not restore the fecundity defects in the H2B quadruple mutants, indicating that H3-H4 chaperone-mediated restoration requires WT H2B. Because cancer cells are unlikely to mutate all the isoforms of their histone genes to oncohistone mutations, inhibiting histone chaperones holds the promise to treat oncohistone-induced cellular abnormalities.

## Discussion

This study demonstrates the advantage of using a genetically tractable system to investigate dysfunctional oncohistone mutations in a living organism. By generating a series of H2B mutant alleles, we found that deleting three of four germline H2B genes caused animal sterility with full penetration, which enabled us to perform forward genetic screens to isolate suppressors. Unexpectedly, we uncovered that inhibiting H3-H4 histone chaperones or reducing H3-H4 histones restored the fertility of *H2B 3KO* animals. Inspired by the genetic results, we provided evidence that the loss of H3-H4 histone chaperones rescued cell division failures in *C. elegans* bearing the dysfunctional oncohistone H2B$^{E74K}$ mutation. Because the loss of histone H3-H4 chaperones caused a reduction of chromatin H3-H4 (Fig 4C–4E), the decreased H3-H4 amount might be comparable to the remaining functional H2A-H2B in chromatin, thereby restoring cell division and fertility. We argue that the loss of histone amount or activity can be tolerated in vivo as long as histone subunits are proportionally reduced.

Consistent with this argument, overexpression of histone subunits caused chromosome loss in diploid budding yeast by perturbing chromatin structure in *Saccharomyces cerevisiae* [18,19,23,24], but co-overexpression of all the core histone subunits reduced the frequency of chromosome loss [18]. Overexpression of histone resulted in embryo lethality and a high

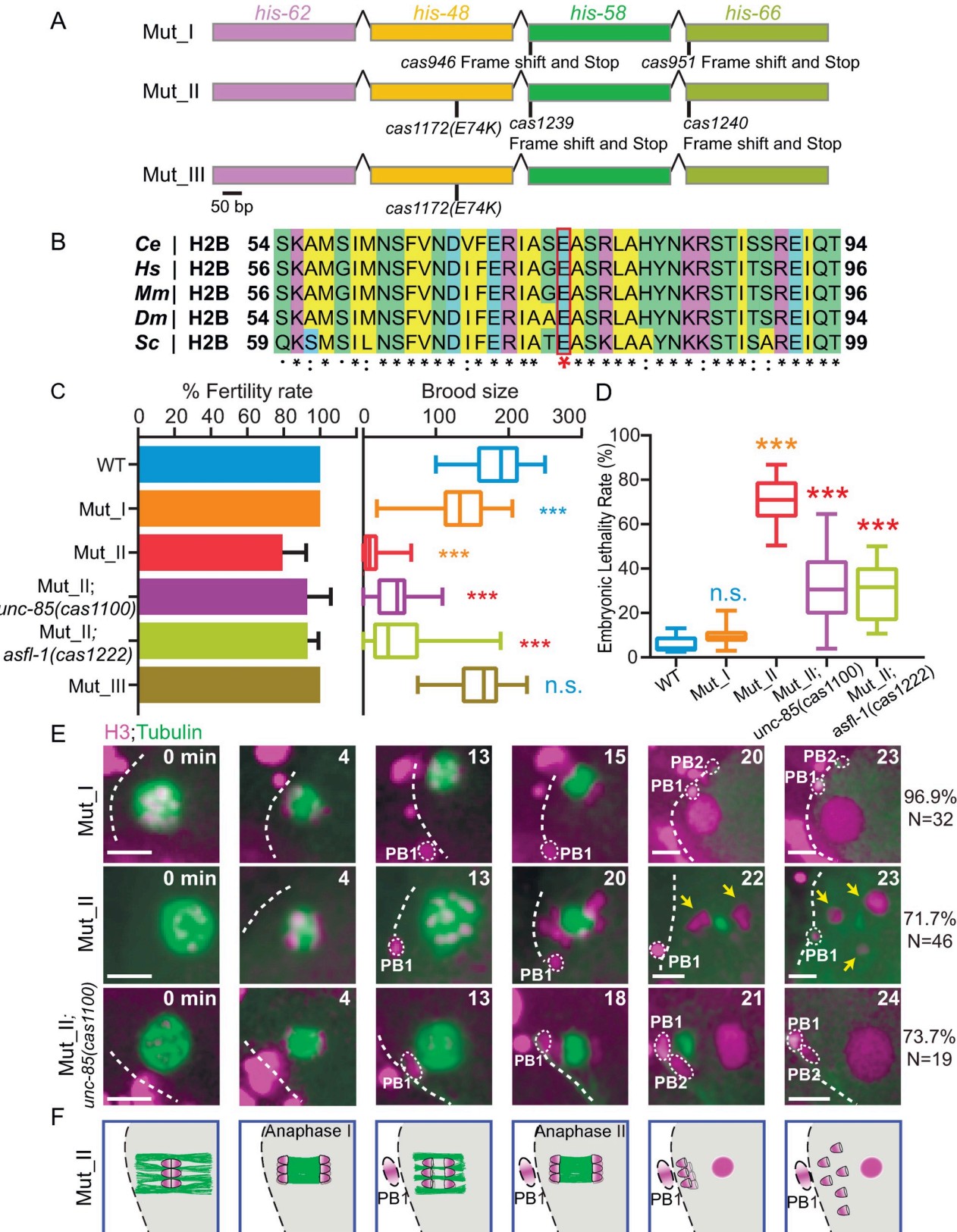

**Fig 5. Inhibition of H3-H4 ASF1 chaperones rescued H2BE74K defects in *C. elegans*. (A)** Schematic of the H2B oncohistone mutant construction. The amino acid changes (*cas1172) in his-48* and the deletion (*cas946 and cas1239*) in *his-58 (cas951 and cas1240) in his-66* mutant alleles are

indicated. Scale bar, 60 bp. **(B)** Multi-sequence alignment of *C. elegans (Ce)* H2B and its homologs in *human (Hs)*, *mouse (Mm)*, *Drosophila (Dm)*, and *Saccharomyces (Sc)*. A red asterisk highlights the oncohistone mutation. **(C)** Fertile adult percentage and brood size of WT, Mut I, Mut II, Mut II; *unc-85(cas1100)*, Mut II; *asfl-1(cas1222)* and Mut III animals at 25˚C. Information of Mut I and Mut II is in (A). N = 9–33, Data are presented as mean ± SD (error bars). Statistical significance compared with the control with a matching color code is based on Student's *t*-test, ***p < 0.001. Three biological replicates calculated the fertile adult percentage. **(D)** Quantification of embryonic lethality rates in Mut I, Mut II, Mut II; *unc-85(cas1100)*, Mut II; *asfl-1(cas1222)* animals at 25˚C. n = 500–1612, from three biological replicates. Statistical significance compared with the control with a matching color code is based on Student's *t*-test, n.s., not significant, ***p < 0.001. **(E)** Fluorescence time-lapse images of GFP-tagged tubulin with mCherry-tagged histone H3 and membrane (magenta) during oocyte meiosis in Mut I, Mut II, and Mut II; *unc-85(cas1100)* at 25˚C. Percentages of meiosis patterns are indicated. Scale bar, 5 μm. **(F)** Enlarged schematic timescales of oocyte meiosis in Mut II.

frequency of chromosome segregation defects in *C. elegans* (S8C and S8D Fig), demonstrating the correct histone subunit ratio's crucial role in animal cells. Biochemical results show that an equimolar ratio of H3-H4 to H2A-H2B is vital for chromatin formation [25]. Recognizing the importance of the balanced histone composition, we suggest that inhibition of H2A-H2B chaperone pathways could suppress the phenotypes caused by dysfunctional H3 and H4 onco-histone mutations. Oncohistone mutations are not ideal for drug development; but our study uncovers that the cellular and animal defects caused by oncohistones can be rescued by manip-ulating a previously unrecognized process independent of oncohistones. We anticipate that the development of small molecules that target the pathways other than the mutated histone genes holds the promise to devise effective strategies for treating oncohistone-induced tumori-genesis. In support of this idea, inhibiting the JNK pathway recovered the aberrant *C. elegans* oocytes resulting from H3K27M-oncohistone mutation and suppressed the abnormal cell rep-licative fate in the human H3.3 K27M tumor cells [26].

## Materials and methods

### *C. elegans* strains, culture, and genetics

All the *C. elegans* strains were genetic derivatives of the wild-type (WT) strain Bristol N2, except for Hawaii strain CB4856 in SNP mapping. Strains were raised on NGM agar plates seeded with *E. coli* strain OP50 at 20˚C following standard procedures [27]. Some strains were obtained from the Caenorhabditis Genetics Center (CGC) funded by the NIH Office of Research Infrastructure Programs (P40 OD010440). Strains in this study are listed in S1 Table.

### Genome editing and molecular biology

We perform genome editing according to established criteria. The sgRNA sequence was designed using the CRISPR design tool (https://zlab.bio/guide-design-resources) and was inserted into the plasmid *pDD162* by linearizing the vector with 15 bp overlapped primers. To make knock-in and mutant strains, 1- to 1.5-kb upstream and downstream homology arms were amplified and inserted into *pPD95.77*. sgRNA construct and homology recombination (HR) template were co-injected into N2 with *pRF4 [rol-6 (su1006)]* and *Podr-1::dsRed* selection markers. Marker-positive F1 animals were singled and screened by PCR and Sanger sequenc-ing [28]. S2 Table lists the sgRNA sequences, and S3 Table lists the primers and plasmids.

### Forward genetic screens

We used forward genetic screens to isolate *H2B 3KO* suppressors. The heterozygous *H2B 3KO* mutant animals (P0) were synchronized at the late L4 larval stage, collected in 4 mL M9 buffer, and incubated with 50 mM ethyl-methanesulfonate (EMS) for 4 hours at room temperature with constant rotation. Animals were then washed with M9 three times and cultured under standard conditions. After 20 hours, adult animals were bleached. Eggs (F1) were distributed

and raised on~100 9 cm NGM plates, each containing 50 to 100 eggs. Adult F2 animals on each plate were collected, and homozygote progenies were segregated among the heterozygotes by fluorescence marker. Homozygote *H2B 3KO* animals were individually cultured to isolate fertile suppressors (S5A Fig). Using the single nucleotide polymorphisms (SNPs), we mapped *cas1100* at -6.93~0.94 cM on chromosome II. We then identified mutations using whole-genome sequencing.

## Germline dissection

Young adult hermaphrodite animals were cut with a syringe at the tail or head region in the gonad dissection buffer to expose their anterior or posterior gonads [29]. The dissected gonads were fixed in -20˚C pre-chilled methanol for 10 min and then transferred to poly-L-lysine coated slides. We rinsed slides with PBS, touched the edges of slides on Kimwipes to remove excess PBS, applied 30 μl mounting medium containing DAPI to the slides, and slowly tipped the coverslip onto the mounting medium to avoid air bubbles, dry at room temperature for 5–10 min and seal the edges of coverslip with nail polish.

## RNA interference and candidates screening

RNAi by feeding was performed as described [30]. In brief, we performed a candidate RNAi screen using *H2B 3KO* worms. The RNAi clones were from the Ahringer *C. elegans* library and confirmed by sanger sequencing. Bacteria were cultured in LB plus carbenicillin at 50 μg ml$^{-1}$ and tetracycline at 12.5 μg ml$^{-1}$ overnight and then seeded on NGM plates containing 50 μg ml$^{-1}$ carbenicillin, 12.5 μg ml$^{-1}$ tetracycline, and 1 mM IPTG. Bacteria containing the empty vector L4440 were used as the control.

## Live-cell imaging

*C. elegans* were anesthetized with 0.1 mmol/L levamisole in M9 buffer for live-cell imaging of meiosis and germline oocyte. We synchronized young adult worms to image early embryos, cut them with syringes to release embryos in M9 buffer, and then mounted them on 2% (wt/vol) agarose pads at 20˚C [31], time-lapse images were taken by μManager (www.micro-manager.org) with an exposure time of 300 ms every 30–60 s. Our imaging system includes an Axio Observer Z1 microscope (Carl Zeiss MicroImaging, Inc.) equipped with a 100X, 1.45 NA objective, an EM CCD camera (Andor iXon+ DU-897D-C00-#BV-500), and the 488-nm and 561-nm lines of a Sapphire CW CDRH USB Laser System to a spinning disk confocal scan head (Yokogawa CSU-X1 Spinning Disk Unit). ImageJ software (NIH) was used to process and quantify the images (http://rsbweb.nih.gov/ij/). The background fluorescence was subtracted for the intensity measurements of UNC-85 and HIS-74 distribution. The whole-cell and nucleus fluorescence intensity of UNC-85 and HIS-74 were measured, and the nucleus to cytoplasm fluorescence ratio (N/C ratio) was calculated to reflect the distribution of UNC-85 and HIS-74.

## Self-brood size assay

Brood size counts were performed by picking individual late L4 stage hermaphrodites to separate plates and transferring daily to new plates until no more eggs were laid. Live progeny were counted at the late larva to adult stage [32].

## Heat-shock treatment and viability

For *C. elegans* embryos viability analysis, 30 young adult worms were collected on a seeded NGM plate to lay approximately 100 eggs. The embryos were heat-shocked for 1 hour at 33˚C

and cultured at 20˚C for three days. Viability is defined as the number of hatched worms divided by the original egg numbers.

## Molecular dynamic simulation and structural analysis

Wild-type yeast Asf1-H3-H4 complex (PDB code: 2HUE) and free Asf1 (PDB code: 1ROC) structures were downloaded from PDB, and G110R mutation structures were created by USCF Chimera (RRID:SCR_004097) with energy minimization. Molecular dynamics simulations were performed using GROMACS (RRID:SCR_014565) version 4.6.7 using the OPLS-AA force field parameter set. After dispersion correction, each system was solvated in 150 mM NaCl solvent with SPC/E water models in a cubic box. Subsequently, neutralizing counter ions were added, and the steepest descent energy minimization was performed, followed by a two-step equilibration consisting of 100 ps of isochoric-isothermal (NVT) equilibration and 100 ps of isothermal-isobaric (NPT) equilibration. All position restraints were removed, and simulations were performed for 5 ns. RMSF and RMSD analysis calculate the standard deviation (root mean square fluctuation/deviation) of the atomic positions of the specified amino acids compared to their starting position within the energy minimized and equilibrated structure. Simulations were performed in triplicate; values shown in figures are averages with a standard deviation of the mean highlighted as error bars. Structural analysis and images were performed and created using PyMol (RRID:SCR_000305). The electrostatic distribution surfaces were generated using the APBS tool (RRID:SCR_008387).

## Chromatin immunoprecipitation

We performed chromatin immunoprecipitation as described previously with some modifications [33]. Synchronized young adults were harvested and washed three times with M9 to remove excess bacteria. Using a 200 µl pipette tip, we dripped the worm mix into a 100 ml beaker that contains liquid nitrogen to make frozen worm "popcorn". Then, we ground the worm "popcorn" into frozen powder with a mortar and crosslinked with 1.1% formaldehyde for 15 min at room temperature. Formaldehyde was quenched by the addition of glycine, and the fixed powder was washed with cold PBS 3 times. The fixed powder was resuspended and homogenized 2–3 strokes using a glass Dounce homogenizer that is assumed to disaggregate cells gently. Nuclei purification and chromatin digestion were conducted using SimpleChIP Enzymatic Chromatin IP Kit (CST, #9003). Briefly, we resuspend cells using buffer A and incubate them on ice for 10 min. Pellet nuclei were collected and resuspended with buffer B. We added the micrococcal nuclease and incubated the reaction for 4–12 min at 37˚C to digest DNA to the length of approximately 150–900 bp. We stopped the digestion by adding EDTA. Then samples were resuspended in ice-cold FA buffer with 0.1% SDS and sonicated using a Bioruptor sonication system at 70% power for 20 cycles of 30 s on and 30 s off. The sonicated samples were then centrifuged at 4˚C for 15 min at 16,000*g*, and we transferred the supernatant to a new tube. The crosslinked chromatin concentration was measured by BCA Protein Assay Kit (TIANGEN, #PA115), and 2% of the extract was saved as the input sample. 5 µg chromatin was precleaned with protein A/G agarose beads. We added 10 ng *Drosophila* line 2 (S2) cells spike-in chromatin (Active Motif, # 53083) to each sample and immunoprecipitated overnight using 2 µg H3 antibody (Abcam, #AB195277) and 0.5 µg spike-in antibody (Active Motif, #61686). We added 30 µl protein A/G agarose beads to each ChIP sample and rotated them at 4˚C for 2 h. We added 150 µl ChIP Elution Buffer and vortex gently at 1200 rpm for 30 min at 65˚C. Then we reversed the crosslink with proteinase K at 65˚C for 2h. DNA was purified using PCR Purification Kit (QIAGEN, #28106), and DNA library construction was conducted with DNA Library Prep Kit for Illumina V3 (Vazyme, #ND607). The library was

sequenced on NovaSeq systems (Illumina) using 150 nucleotides kits and paired-end sequencing (Novogene).

### ChIP-seq data analysis

Raw reads were filtered using cutadapt software (version 2.5) for clean reads and checked with fastqc software (version 0.11.5). Clean reads were aligned to a mixed reference worm genome using BWA (version 0.7.17). The reference genome was mixed in advance with *Caenorhabditis elegans* DNA toplevel assembly (WBcel235 from the ensemble) and *Drosophila melanogaster* DNA toplevel assembly (BDGP6.32 from the ensemble). The aligned reads were separated into two sequence-alignment map (SAM) files. The SAM file of *Caenorhabditis elegans* was then converted into binary alignment map (BAM) files with samtools software (version 1.9) for further analysis, and the SAM file of *Drosophila melanogaster* was used for calculating spike-in factors [34]. The sorted and indexed BAM file of *Caenorhabditis elegans* underwent filtration using sambamba software (version 0.6.6) with 'view -h -f bam -F "not unmapped and not duplicate" '. The filtered BAM file was then converted to bigwig(bw) file using deeptools software (version 3.3.1) with 'bamCoverage--binSize 10--scaleFactor **α**--effectiveGenomeSize 100286401--extendReads--ignoreDuplicates--bam 'where **α** stood for the Spike-in factors calculated from SAM file of Drosophila melanogaster [35]. The Spike-in factors were calculated as David A. described with this formula [17]:

$$Spike-in\ factor = \frac{1}{All\ mapped\ reads}$$

We generated heatmaps and profile maps of Histone H3 occupation using deeptools software (version 3.3.1). We used Integrative Genomics Viewer (IGV, version 2.7.2) to visualize signal tracks to view bigwig files generated from deeptools software [36]. The comparison heatmaps in IGV between samples were generated using 'bigwigCompare 'in deeptools software.

### Supporting information

**S1 Fig. Genomic loci of the *C. elegans* histone genes. Related to Fig 1. (A)** Multiple protein sequence alignment of five *C. elegans* (Ce) H2B proteins. An asterisk (\*) indicates positions 100% conserved in the alignment, a colon (:) indicates conservation between groups of strongly similar properties. **(B)** Genomic loci of all histone genes in *C. elegans*. Yellow boxes indicate H2A, red boxes indicate H2B, blue boxes indicate H3, and green boxes indicate H4. (TIF)

**S2 Fig. The complete inventory of the endogenously GFP-tagged *C. elegans* H2B knock-in embryos during metaphase at the one-cell stage. Related to Fig 1.** The HIS-58 subfamily H2B expresses in one-cell stage embryos. Embryo fluorescence was visualized by *GFP::H2B* (green) and *Ppie-1::H2B::mCherry* (magenta). Scale bar, 5 μm. (TIF)

**S3 Fig. The complete inventory of the endogenously GFP-tagged *C. elegans* H2B knock-in germlines. Related to Fig 1.** The HIS-58 subfamily H2B expresses in *C. elegans* germlines. Germline fluorescence was visualized by *GFP::H2B* (green) and *Ppie-1::H2B::mCherry* (magenta). Scale bar, 10 μm. (TIF)

**S4 Fig. H2B depletion in the germline causes sterility of *C. elegans*. Related to Fig 1. (A)**
Histone gene clusters on chromosome IV. Magenta arrows indicate four germline H2B genes.
**(B)** DNA sequences of germline H2B mutants generated via CRISPR-Cas9. The 20 nt target
sequences and the following PAMs (NGG) are highlighted in green and magenta, separately.
Purple sequence, PCR primers; Red dashed lines, deleted nucleotides. **(C)** Immunofluores-
cence images with the anti-H2B antibody of WT, *H2B 3KO*, and *H2B 3KO; unc-85(cas1100)*
quadruple mutant embryos. DAPI stained nuclei. Scale bar, 5 μm. **(D)** Quantification of his-
tone H2B fluorescence intensity ratio relative to DAPI of WT, *H2B 3KO*, and *H2B 3KO; unc-
85(cas1100)* embryos. N = 28–43. Data are presented as mean ± SD. Statistical significance
compared with the control with a matching color code is based on Student's *t*-test, n.s., not sig-
nificant, ***p < 0.001.
(TIF)

**S5 Fig. Molecular dynamic simulation indicated the G110R mutation-induced disruption
of Asf1-histone binding. Related to Figs 1 and 2. (A)** A flowchart of the suppressor screen-
ing. *H2B 3KO* animals are sterile (Ste). At the late L4 stage, animals were mutagenized with
ethyl methanesulfonate (EMS). Viable F3 animals were candidate suppressors. The mutant
genes were cloned using single-nucleotide polymorphism mapping combined with whole-
genome sequencing. **(B)** Superposition of free and histone-bound Asf1 structures with the
Asf1-H3 binding interface highlighted. The mutation site (G110R) within the H3-binding cleft
of Asf1 is situated closely to the positively charged H3 surface in the Asf1-H3-H4 complex
structure. **(C)** Molecular dynamic (MD) simulation of Asf1-H3-H4 complex suggests the
G110R-induced dissociation of H3-H4 from Asf1. G110R introduced a positively charged sur-
face bulge within the H3-docking pocket, and as a result, the α3 helix of H3 responsible for
Asf1 binding underwent a significant deflection away from Asf1 shown in the time-lapse
images of the simulation of mutant structure. **(D)** Statistical analysis of the deflection angles of
the α3 helix in the MD analysis. **(E)** Structural representation of the Asf1-H4 interface showing
the rotation of Asf1 R145 required for H4 binding. **(F)** Asf1 G110R structure shows an increase
of movement within the β8–9 loop, including R145, which appears to hinder the H4 binding.
Time-lapse images highlight the changes in the β8–9 loop throughout the simulation. **(G)** Sta-
tistical analysis of the structure deviation of residues in the β8–9 loop in the simulation. Values
are representative of three independent simulations, and error bars indicate mean ± SD,
*p<0.05, ***p<0.001. Asf1-H3/H4 complex (PDB code: 2HUE) and free Asf1 (PDB: 1ROC)
structures were used for MD and structural analysis.
(TIF)

**S6 Fig. UNC-85 mutations rescue the sterility of *H2B 3KO* animals. Related to Fig 2. (A)**
Fluorescence time-lapse images (up) and heatmaps (down) of UNC-85::GFP (left) and UNC-
85::GFPG111R (right) in the germline at the day one adult stage. GFP images are shown as
heatmaps below the merged images. Scale bar, 5 μm. **(B)** Quantifications of GFP-tagged UNC-
85 total fluorescence intensity in -1 and -2 oocyte (left) and intensity ratio of the nucleus to the
cytoplasm (right). Data are presented as mean ± SD (error bars). Statistical significance based
on two-way ANOVA, n.s., not significant, ***p < 0.001. N = 57–117.
(TIF)

**S7 Fig. Reduced histone H3 in nucleus and chromatin in the *unc-85* mutant embryos.
Related to Fig 4. (A)** Fluorescence time-lapse images of GFP-tagged HIS-74 with mCherry-
tagged (magenta) membrane of the germline in WT (left) or *H2B 3KO; unc-85(cas1100)* qua-
druple mutant (right) at the day-one adult stage. GFP images are shown as heatmaps below
merged images. Scale bar, 5 μm. **(B)** Quantifications of GFP-tagged HIS-74 total fluorescence

intensity in -1 and -2 oocyte (left) and intensity ratio of the nucleus to the cytoplasm (right). Data are presented as mean ± SD (error bars), statistical significance based on two-way ANOVA, n.s., not significant, ***p < 0.001. N = 37–52. **(C)** Genome tracks show H3 signals of a representative genomic locus (Chr IV: 10,627,008–10,840,892) in WT (up) and *H2B 3KO; unc-85(cas1100)* quadruple mutant animals (middle). Heatmap shows H3 depletion of *H2B 3KO; unc-85(cas1100)* quadruple mutant relative to WT (bottom). Dark blue boxes show the reference genes. **(D)** Immunofluorescence images with the anti-H4 antibody of WT, *H2B 3KO*, and *H2B 3KO; unc-85(cas1100)* quadruple mutant embryos. DAPI stained nuclei. Scale bar, 5 μm. **(E)** Quantification of histone H4 fluorescence intensity ratio relative to DAPI of WT, *H2B 3KO*, and *H2B 3KO; unc-85(cas1100)* quadruple mutant embryos. N = 43–101. Statistical significance compared with the control with a matching color code is based on Student's *t*-test, n.s., not significant, ***p < 0.001.
(TIF)

**S8 Fig. Overexpression of H2B and H3 induced defective chromosome segregation during mitosis. Related to Figs 4 and 5. (A)** Schematic of *hsp16.41::his-48::gfp* single-copy insertion using the CRISPR-Cas9-based genome editing strategy. Scissor shows the Cas9 cleavage site, with the single-guide RNA (sgRNA) target sequence above and the PAM (Protospacer Adjacent Motif) sequence in red. A 5.6 kb homology arm was inserted into the plasmid pPD95.77 to make the homology recombination (HR) template. **(B)** Quantification of embryonic lethality rates in WT, H2B, and H3 overexpression alleles. Data are presented as mean ± SD (error bars). Statistical significance based on Student's *t*-test, ***p < 0.001. N = 57–117 for H2B overexpression, N = 413–1242 for H3 overexpression. Data were from three biological replicates. **(C)** Fluorescence time-lapse images of GFP-tagged H2B during mitosis in WT and H2B overexpression embryos. Metaphase was defined as time zero. Scale bar, 5 μm. **(D)** Fluorescence time-lapse images of GFP-tagged H3 during mitosis in WT and H3 overexpression embryos. Metaphase was defined as time zero. Arrow indicates anaphase bridges, and arrowheads indicate the lagging chromatin. Percentages of meiosis patterns were indicated. Scale bar, 5 μm. **(E)** Fluorescence images of GFP-tagged tubulin with mCherry-tagged histone H3 and membrane (magenta) of one cell stage embryo in Mut I and Mut II at 25˚C during pronucleus meet and first mitosis. Orange arrows show pronucleus from sperm, and yellow arrows show pronucleus from the oocyte.
(TIF)

**S1 Video. UNC-85 mutations impair UNC-85's nuclear localization in vivo. Related to Fig 2.** Fluorescence time-lapse heatmaps of UNC-85::GFP (left) and UNC-85::GFP[G111R] (right) in the one-cell stage *C. elegans* embryo. The first occurrence of the pronucleus meeting was defined as time 0 min, and frames were taken every 1 min.
(AVI)

**S2 Video. H2B depletion in the germline results in the sterility of *C. elegans*. Related to Fig 2.** In utero fluorescence time-lapse images of meiosis visualized by GFP::tubulin (green) and mCherry::Histone (magenta) in wild type (left) and *H2B 3KO* (right). Metaphase I was defined as time zero, and frames were taken every 1 min.
(AVI)

**S3 Video. UNC-85 mutations rescued sterility in *H2B 3KO*. Related to Fig 1.** In utero fluorescence time-lapse images of meiosis visualized by GFP::tubulin (green) and mCherry::Histone (magenta) in *H2B 3KO; cas1100* quadruple mutant. Metaphase I was defined as time zero, and frames were taken every 1 min.
(AVI)

**S4 Video. UNC-85 mutation impairs histone H3 HIS-74 nuclear localization in vivo. Related to Fig 4.** Fluorescence time-lapse heatmaps of HIS-74::GFP in the one-cell stage *C. elegans* embryo in WT (left) and *H2B 3KO; cas1100* quadruple mutant (right). The first occurrence of the pronucleus meeting was defined as time 0 min, and frames were taken every 1 min.
(AVI)

**S5 Video. H2B oncohistone mutant caused meiosis defects in *C. elegans*. Related to Fig 5.** In utero fluorescence time-lapse images of meiosis visualized by GFP::tubulin (green) and mCherry::Histone (magenta) in control Mut I (left) and H2B oncohistone mutant Mut II (right). Metaphase I was defined as time zero, and frames were taken every 1 min.
(AVI)

**S6 Video. Meiosis defects in H2B oncohistone mutant were suppressed by an *unc-85* mutant. Related to Fig 5.** In utero fluorescence time-lapse images of meiosis visualized by GFP::tubulin (green) and mCherry::Histone (magenta) in Mut II; *cas1100* double mutant. Metaphase I was defined as time zero, and frames were taken every 1 min.
(AVI)

**S1 Table. *C. elegans* Strains in this study.**
(DOCX)

**S2 Table. CRISPR-Cas9 Targets in this study.**
(DOCX)

**S3 Table. Plasmids and Primers in this study.**
(DOCX)

## Acknowledgments

We thank Drs. G. Li, Q. Li, B. Zhu, and Z. Chen for discussion.

## Author Contributions

**Conceptualization:** Ruixue Zhao, Guangshuo Ou.

**Formal analysis:** Ruixue Zhao, Ruxu Geng, Guangshuo Ou.

**Funding acquisition:** Guangshuo Ou.

**Investigation:** Ruixue Zhao, Zhiwen Zhu, Ruxu Geng, Xuguang Jiang.

**Supervision:** Guangshuo Ou.

**Validation:** Ruixue Zhao, Guangshuo Ou.

**Writing – original draft:** Ruixue Zhao, Guangshuo Ou.

**Writing – review & editing:** Ruixue Zhao, Wei Li, Guangshuo Ou.

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
