## [Decision Letter · Decision Letter 0]

21 Mar 2022

Dear Dr Zhao,

Thank you very much for submitting your Research Article entitled 'Inhibition of Histone H3-H4 Chaperone Pathways Rescues Dysfunctional Oncohistone H2B' to PLOS Genetics.

The manuscript was fully evaluated at the editorial level and by independent peer reviewers. The reviewers appreciated the attention to an important problem, but raised some substantial concerns about the current manuscript.

As you will see from the critiques, all 3 reviewers expressed interest in the manuscript, particularly your conclusion that the overall stoichiometry of histones within nucleosomes, and in chromatin overall, is more important than the actual abundances of any of the four different histone dimers or tetramers. The results presented from the C. elegans studies raised only minor concerns, mostly with issues related to the description of the results and the writing in some places, as nicely detailed by Reviewer 2, and I agree that the conclusions from this part of the manuscript are solid, intriguing, and likely to be of interest to others working in the chromatin field.

The major concerns from all 3 reviewers center on the work in the human cells. I agree that this section of the study needs significant strengthening, or absent that, be removed from the manuscript. Please review carefully the criticisms of this section and the additional experiments that the reviewers suggest will be necessary to strengthen these results. The title of any resubmission should also reflect the major conclusions from the work, which if the human cell studies are removed will necessitate a change in the title.

Based on the reviews, we will not be able to accept this version of the manuscript, but we would be willing to review a much-revised version. We cannot, of course, promise publication at that time.

If you decide to revise the manuscript for further consideration at PLOS Genetics, please aim to resubmit within the next 60 days, unless it will take extra time to address the concerns of the reviewers, in which case we would appreciate an expected resubmission date by email to plosgenetics@plos.org.

[LINK]

We are sorry that we cannot be more positive about your manuscript at this stage. Please do not hesitate to contact us if you have any concerns or questions.

Yours sincerely,

William G. Kelly

Guest Editor

PLOS Genetics

Gregory P. Copenhaver

Editor-in-Chief

PLOS Genetics

Reviewer's Responses to Questions

**Comments to the Authors:**

Reviewer #1: In this manuscript, Zhao et al use C. elegans to investigate the consequences of H2B loss in the germline and early embryos. First and perhaps not surprisingly, they find that H2B triple KO of 3/4 paralogues that are expressed in these cells results in sterility. Interestingly, they then perform suppressor screens and identify that loss of H3-H4 chaperones can restore fecundity. A particularly intriguing finding is that knockdown HIRA or ATRX/DAXX both restore fecundity even though they are responsible for H3.3 deposition in distinct types of chromatin, and that CAF-1 knockdown, which is responsible for replication-coupled deposition primarily of H3.1, also restores knockdown. Thus, impairing any H3 deposition pathway rescues this phenotype. They identify defects in meiosis in the H2B 3KO worms, and that H3 chaperone knockdown restores proper chromosome segregation during meiosis. Finally, in an attempt to link their H2B KO findings to human disease, they examine H2BE76K (74K in worms) effects on fertility and find that only in the context of the H2B 2KO does the H2B-E74K have an effect, which is more modest than the 3KO. They also express H2B-E74K in HEK293 cells and find a subtle mitotic delay phenotype. The C. elegans work and cleanliness of the system, as well as simplicity of these experiments, are convincing in demonstrating both that loss of H2B causes sterility and that H3/H3 chaperone knockdown rescues this phenotype. The work in 293 cells is more complicated and the relevance to their C. elegans findings is less clear. This is in part because in the 293 context the mutant histone is overexpressed and likely a small fraction of the total H2B, whereas in the worm experiments H2B-E74K comprises a much larger fraction of the total H2B pool in the 2KO background. Further, the 293 phenotype is very mild and it appears that the variance in MI is more impacted than the number of cells in mitosis. Finally, 293 cells have a high rate of MI at baseline. The link between H2B KO and H2B-E74K is also somewhat more complicated because H2B-E76K destabilizes nucleosomes, which is different than not making H2B in the first place. However, the screen results are fascinating and I think this work stands alone without the human experiments, but if those are to be included they should be strengthened.

Minor Points:

1. If intending to include the human work, an orthogonal assay (e.g. flow cytometry w/DAPI) to examine cell cycle progression in H2B-E76K expressing 293T cells to show whether cells are accumulating in M. Can be coupled with H3S10 or S28 staining as is done in commercially available assays. The DAPI result is important because the Asf-1 rescue could be explained by less H3S10 deposition in chromatin as a result of less H2B deposition in chromatin.

2. Western blotting for H2B-E76K expression in 293 should be done (I presume the constructs are tagged) and anti-H2B WB should be used to assess expression of the mutant (or WT control) vs total H2B pool.

3. Not essential, but can the authors stain for CENP-A and count foci in 293 cells to determine whether deposition is impacted? This might explain the mitotic phenotype although HJURP is the CENP-A chaperone, not Asf-1.

4. The word "chaperone" is misspelled several times, particularly in the paragraphs the data in figure 5.

Reviewer #2: The manuscript titled “Inhibition of Histone H3-H4 Chaperone Pathways Rescues Dysfunctional Oncohistone H2B” by Guangshuo Ou and co-authors provides an important contribution to our understanding of chromatin dynamics in animals.

The key conclusions of the authors are well supported by the presented data. Specifically, they establish in an unbiased manner that sterility of C. elegans lacking several H2B genes is not due to the histone/nucleosome depletion per se, but is due to the disruption of stoichiometry between H2A/H2B dimers and H3-H4 tetramers. Furthermore, the authors model oncogenic H2B (E76K) mutation that destabilizes nucleosome core in C. elegans and show the potential of H3-H4 chaperone loss in rescuing the sterility phenotype. In parallel, it is shown that H2B (E76K)-driven mitotic arrest in human cell culture can be rescued by H3-H4 chaperone downregulation. The latter result suggest a possibility of using histone chaperone inhibitors for treatment of some cancers caused by oncohistones.

Scientifically, I support publication of the manuscript. I have concerns regarding manuscript language that often affects meaning and some other suggestions/corrections.

I was not able to evaluate GEO submission without the token number.

Specific comments:

1. Title. The title only reflects the oncohistone part of the story, a more general one referring to the importance of H2A/H2B and H3-H4 balance may work better.

2. Summary has to be edited for word omission. Also, “fertility” works better than “animal defects”.

3. Introduction. The contrast between the statements in sentence 3 “The best-….” and sentence 4 “In contrast, ….” is not clear. Did the authors mean to contrast oncomutations that disrupt proper histone tail PTMs with those affecting nucleosome structure?

4. “facilitated us” has to be changed

5. Page 6: Here and throughout the manuscript “loss of UNC-85 restored sterility in H2B 3KO animals”; change to “restored fertility” or “rescued the sterility”, also “rescue meiosis defects”.

6. Page 8: change “reduced histone H3 proteins from chromatin”

7. Methods. Molecular dynamic simulation. Can this section be described in a more comprehendible manner for readers like me? “protein constructs were placed in a cubic box of 100 nM NaCl in simple point charge water” reads like “wet lab” experiment description. Change to “protein models”? “simulation of a cubic box with 100 nM NaCl”?

8. Methods. Chromatin Immunoprecipitation. Change “grind worms into frozen powder….” sentence.

9. Methods. Chromatin Immunoprecipitation. “We reversed the crosslink with Proteinase K”. Formaldehyde crosslinking is reversed by heat, i.e. 30 min at 65C.

10. References. #18: First names of the authors are listed instead of their last names. Douglas Meeks-Wagner and Leland H.Hartwell

11. Figure 1 legend. (B) include “endogenously GFP-tagged”, (D) “proteins” plural, (E) only a portion of protein sequences is aligned; what is the significance of this domain?

12. Figure 2 (F). Provide the number of oocytes scored, and number of animals used.

13. Include “Statistical significance compared with the control with a matching

color code” explanation in legends to Figures 3, 5, etc.

14. Figure 4D, I do not understand the legend. The Y axis is labeled “genes”, is this correct. The “gene distance bp” on the x-axis can correspond to the 3Kb flanks of genes, not to the gene model.

14. Figure 4E. I suggest replacing “H3 enrichment” with “H3 depletion” to better convey the point, same for relevant supplemental figures.

15. Figure S3. Indicate that DNA is shown in magenta, since on figure S2 histone H2B is shown in this color.

16. Figure S5. I do not think panels G, I, J are necessary.

17. Figure S6. Change to “rescue sterility”

Reviewer #3: In this manuscript, the authors report that by blocking H3-H4 chaperones like UNC-85, ASFL-1 and others, they were able to restore defects in H2B 3KO C. elegans, including cell division and fertility defects. Then they extended to the claim that the inhibition of H3-H4 chaperones largely restore H2BE76K-induced defects in both C. elegans and HEK293 cells. However, the developmental defects are more likely caused by insufficient H2B supply rather that E74K mutant in C. elegans.

The human cell experiments appear to be not as solid and I suggest to remove them or to provide substantial amounts of supporting evidence.

Major points:

1. HIRA and DAXX complexes are responsible for depositing H3.3 into transcriptional active regions and telomeric/centromeric regions, respectively. On the other hand, CAF-1 is required for replication-coupled nucleosome assembly. Why knockdown of all these chaperones with distinct functions sucessfully rescued H2B 3KO defects? Any explanations?

2. In humans, H2B is coded by dozens of genes, and histones are highly abundant in human cells. It is hard to imagine that severe mitotic defects (56%, Fig. 5E) can be caused by the overexpressed H2B E74K, unless the mutation has a very strong dominant effect. If that is the case, cas1172(E74K) alone should lead to brood size defects. What is the phenotype when E74K occurs on only one histone H2 protein (like his-48E74K, not at the background of his-58/his-66 dKO)?

3. Fig. S8A-8D, the authors used heat-shock system to express histones in the embryos. However, the level of overexpression is not determined. In addtion, a negative control that only overexpresses GFP is missing.

Minor points:

1. Third line of figure legend in Fig. S7, it is suggested to use quadruple KO or quadruple mutant instead of "double mutant", which may cause confusion about the genotype of the animal.

2. KD efficiency of Asf1a and Asf1b in HEK293T cells should be verified.

3. In Fig 5C and 5D, a WT control should be added.

**Have all data underlying the figures and results presented in the manuscript been provided?**

Reviewer #1: Yes

Reviewer #2: Yes

Reviewer #3: Yes

PLOS authors have the option to publish the peer review history of their article (what does this mean?). If published, this will include your full peer review and any attached files.

Reviewer #1: No

Reviewer #2: No

Reviewer #3: No

---

## [Decision Letter · Decision Letter 1]

28 Apr 2022

Dear Dr Zhao,

We are pleased to inform you that your manuscript entitled "Inhibition of Histone H3-H4 Chaperone Pathways Rescues C. elegans Sterility by H2B Loss" has been editorially accepted for publication in PLOS Genetics. Congratulations!

There are a couple typos (see below) that you can address as you prepare your final draft for the production team (the editorial team will not need to re-evaluate).

Yours sincerely,

William G. Kelly

Guest Editor

PLOS Genetics

Gregory P. Copenhaver

Editor-in-Chief

PLOS Genetics

Comments from the reviewers (if applicable):

All reviewers agree that this revised version addresses all of their concerns about the original submission, and also agree that this is an important study and highly suitable for publication in PLoS Genetics. One reviewer suggested the following minor edits:

There is a typo on line 246 "overpressing"

One subtitle (lines 253, 254) still requires a correction. I suggest: "Inhibition of H3-H4 chaperones rescues abnormalities caused by H2B oncohistone in C. elegans"

Reviewer's Responses to Questions

**Comments to the Authors:**

Reviewer #1: By removing the 293T experiments and making the requested textual changes, the authors have appropriately addressed my comments.

Reviewer #2: I am largely satisfied with the revised version of the manuscript and recommend it for publication.

Minor comments:

There is a typo on line 246 "overpressing"

One subtitle (lines 253, 254) still requires a correction. I suggest: "Inhibition of H3-H4 chaperones rescues abnormalities caused by H2B oncohistone in C. elegans"

Reviewer #3: I have no further concerns.

**Have all data underlying the figures and results presented in the manuscript been provided?**

Reviewer #1: Yes

Reviewer #2: Yes

Reviewer #3: Yes

PLOS authors have the option to publish the peer review history of their article (what does this mean?). If published, this will include your full peer review and any attached files.

Reviewer #1: No

Reviewer #2: No

Reviewer #3: No

**Data Deposition**

http://datadryad.org/submit?journalID=pgenetics&manu=PGENETICS-D-22-00207R1

**Press Queries**

---

## [Editor Report · Acceptance letter]

26 May 2022

PGENETICS-D-22-00207R1 

Inhibition of Histone H3-H4 Chaperone Pathways Rescues *C. elegans* Sterility by H2B Loss 

Dear Dr Ou, 

We are pleased to inform you that your manuscript entitled "Inhibition of Histone H3-H4 Chaperone Pathways Rescues *C. elegans* Sterility by H2B Loss" has been formally accepted for publication in PLOS Genetics! Your manuscript is now with our production department and you will be notified of the publication date in due course.

With kind regards,

Livia Horvath

PLOS Genetics

On behalf of:
